# Experience of Late Miscarriage and Practical Implications for Post-Natal Health Care: Qualitative Study

**DOI:** 10.3390/healthcare10010079

**Published:** 2022-01-01

**Authors:** Milda Kukulskienė, Nida Žemaitienė

**Affiliations:** Department of Health Psychology, Faculty of Public Health, Lithuanian University of Health Sciences, 44307 Kaunas, Lithuania; nida.zemaitiene@lsmuni.lt

**Keywords:** miscarriage, spontaneous abortion, reproductive health, women’s wellbeing, qualitative research, thematic analysis, phenomenology

## Abstract

Miscarriage is the most common reason for pregnancy loss, affecting around one in four pregnancies. It is classified as a traumatic event, associated with an increased risk for depression, anxiety, post-traumatic stress, alcohol dependence, somatic symptoms, sexual dysfunction, suicide, and complicated grief. This study aimed to analyse experiences of late miscarriage and to describe practical implications for post-natal health care based on characteristics of pregnancy loss revealed in a qualitative study. Seven women who had late miscarriages participated in in-depth biographical interviews. A phenomenological thematic analysis was applied. Experiential characteristics of late miscarriage were described by four themes and 13 subthemes: the initial splitting state (Dissociation, An Opened Void, An impaired Symbiosis, and The Body is Still Pregnant while the Psyche is Mourning); Betrayal of the body (Symbolic Experience of Internalised Death, Shocking Materiality of the Ongoing Miscarriage, Lost control of the Body, and Confusing Body Signals); Disconnecting (Depersonalising Medical Environment, Guilt Falsifies perception, and Retreat as a means of Self-Preservation); and Reconnecting (Collecting Shatters and Reinterpretation of Maternal Identity). Based on the results of the experiential analysis, another four themes represent practical recommendations for post-natal health care: Informing, Opportunity for Goodbye, Attention to Emotional Wellbeing, and Respectful Hospital Environment.

## 1. Introduction

Miscarriage or spontaneous abortion is the most common reason for pregnancy loss, affecting approximately one in four pregnancies [1,2]. The definition of miscarriage varies among the world, though miscarriage can be generally defined as the loss of a pregnancy before foetal viability [3]. The WHO definition specifies the term of miscarriage up to 22 gestational weeks, while, in the UK, the defined limit is 24 completed weeks of gestation and, in the US, 20 completed weeks’ of gestation [4,5,6,7]. Miscarriages can be divided into early (<10–14 weeks) and late (>10–14 weeks), preclinical and clinical, incomplete and complete, spontaneous or induced, and single or recurrent (in the case of three or more miscarriages) [3,8,9,10,11].

Miscarriage is likely to have adverse effects on physical and mental health [12,13,14]. It is classified as a traumatic event, associated with an increased risk for depression, anxiety, post-traumatic stress, alcohol dependence, somatic symptoms, sexual dysfunction, suicide, and complicated grief [12,13,14,15,16,17,18,19]. It is indicated that women experience high levels of posttraumatic stress, anxiety, and depression after pregnancy loss, and distress tends to remain at clinically important levels up to 9 months [20]. In the case of the pregnancy loss in the second trimester, the level of post-traumatic stress symptomatology 6 weeks after termination was found to be significantly higher compared to a first trimester pregnancy loss [21].

Research shows that a sizable percentage of people seem to experience a grief reaction following miscarriage [22]. Furthermore, experiences of grief might be specific from other losses, as miscarriage is a bodily experience associated with an experienced physical pain, emptiness, guilt, shame, and stigma, as well as the sense of losing part of the self and a personal identity [23,24]. Complicated grief is found to be particularly high in patients after an induced miscarriage due to the diagnosed foetal pathology, which is often a case of a second trimester pregnancy loss [25].

This reproductive health problem is complex, as it affects the whole family [26]. Comparative studies have shown that some partners also report clinically significant levels of PTS, anxiety, and depression after miscarriage, though to a lesser extent than women [27]. It is known that perinatal loss can have negative implications on a partner’s psychological and social wellbeing, which might be related to their attachment to the pregnancy [28]. Furthermore, previous foetal losses can present a long-term emotional and relational challenges to a family and be associated with psychological difficulties during the next pregnancy and parenthood [29,30]. However, psychological and physical effects of miscarriage are commonly underappreciated [31]. Although a number of quantitative studies have been conducted, their results are quite fragmented and lack a holistic theoretical model to explain the process of survival and coping after a miscarriage. There is a lack of information on the vulnerability and resilience factors, which would allow to foresee the predictors of physical and mental health disorders. Moreover, there is a need of more in-depth information on the different situational aspects of miscarriage, as most studies analyse early pregnancy losses. Finally, health professionals working in post-natal health care lack practical guidance on how to provide emotional support to women and their partners after miscarriage.

This study aimed to analyse experiences of late miscarriage and to describe the practical implications in a health care system based on the revealed qualitative characteristics of pregnancy loss. Instead of deductive hypothesis, several inductive research questions are raised: (1) What are the most important experiential characteristics of late miscarriage phenomenon in women? (2) Which psychological needs should be prioritised after a late miscarriage? (3) What practical implications for post-natal health care could be suggested?

## 2. Materials and Methods

### 2.1. Procedure and Participants

A phenomenological qualitative research strategy was chosen. Seven women who had one or more late miscarriages (>12 gestational weeks of pregnancy) participated in the study (Table 1). Purposive “snowball” sampling was applied. The researchers shared invitation advertisements in different social media pages, groups, and blogs. Twenty-four women replied to the invitation (17 of them had experienced one or more early miscarriages and 7 one or more late miscarriages). The inclusion criteria of this study were: female gender, late miscarriage experienced more than 3 months ago, the onset of miscarriage was spontaneous or missed miscarriage was diagnosed at a gynaecologist’s visit, and the participant voluntarily expressed a desire to participate in the interview. The miscarriages experienced by participants varied by type: cases of spontaneous-complete miscarriage, spontaneous-partial miscarriage, medically induced miscarriage, and surgical interventions were included. All of the participants had one or more hospital-assisted late miscarriage, as medical interventions were necessary because of the complexity of late miscarriage. All of the participants were heterosexual, married, or lived with their partners. No additional demographic information was collected to protect the anonymity of the participants.

The procedures for data collection and storage were approved by the Kaunas Regional Biomedical Research Ethics Committee (No. BE-2-99, 23 October 2019). Data was collected in 2019–2020, applying individual in-depth interviews (~50–80 min in length). The interviewer did not personally know or did not have a previous relationship with any of the participants. The interview questionnaire consisted of a main biographical research question: “I would like to hear your story about the miscarriage experienced. Please, feel free to tell me what you want, in the way you want to. Tell me what you remember and what comes to your mind about that period. So please, remember your story and tell me about your miscarriage.” Additional questions about support provided and lacked support were also asked. Semantically and latently important keywords were highlighted, and on this basis, prompting questions were asked. Locations of interviews were Lithuanian University of Health Sciences (psychological counselling room), Vilnius National Martynas Mažvydas Library (private meeting room), “Nerimo Klinika” (private psychotherapeutic cabinet), private office, and participant’s home. In-depth interviews were conducted by a PhD student with a master’s degree in Health Psychology. If there was a need, each participant was provided with information about the possibility of psychological help in their city and directed to professional help in the case of psychological crisis.

### 2.2. Data Analysis

A phenomenological thematic analysis was applied. Thematic analysis (TA) is a widely used, flexible, well-known method in qualitative research, intended for identifying, analysing, and reporting patterns within data, which minimally organises and describes a data set in rich detail [32,33]. Furthermore, TA provides an opportunity to interpret data and explain it more in-depth [34]. TA is a relatively unique analysis strategy in qualitative research, as it only provides a method for data analysis but does not prescribe methods of data collection, theoretical positions, or epistemological and theoretical frameworks [33]. In order to have an epistemological basis, several principles of descriptive phenomenological analysis were relied upon: “bracketing”, descriptive and in-depth open coding, and translating meanings to psychological language [35,36]. Themes were identified in a data-driven inductive way. Developing a saturated phenomenon description was sought. A qualitative paradigm, according to the methodology authors, uses qualitative data and is the analysis of words that are not reducible to numbers, so the number of repetitions was not included in the results [33]. Data analysis was obtained by applying 6 data analysis phases according to V. Braun and V. Clarke (2006) [32]:Familiarising with data. Transcribing the data, mindfully reading the transcript several times, and noting initial insights in the researcher’s diary.Generating initial codes. Open inductive coding was applied. The data were divided into segments according to the changes in the semantic and latent contents, e.g., “Taking full responsibility for the miscarriage”. All the relevant features of the data were coded across the entire data set. Searching for themes. Recurring codes were collated into potential thematic units, e.g., codes such as “Self-destruct”, “Search for a clear cause of miscarriage in one’s behaviour”, “Self-aggression”, and “Taking full responsibility for the miscarriage” constitute subtheme “Guilt distorts perception”. For each subtheme to be included in the results, it had to be mentioned by more than half of the participants (4 or more). All of the main themes were revealed by all of the 7 participants.Reviewing primary themes. Themes were checked, and changes in the formulation and meaning were made, depending on the primary code and the quote. The thematic model was generated. Defining and naming themes. Specifics and definitions of each theme were purified. Illustrative quotes and key features of each theme were described in a separate table. The order of the themes was laid out. The validity criteria for external heterogeneity and internal homogeneity were applied. The thematic model was validated with methodological experts.Producing the report. Clear, vivid, and the most substantive quotations were selected for the illustration of the analysis. The structure of the thematic model was substantiated, and a written report of the analysis was produced [32].

## 3. Results

The results of the study are revealed in eight themes and are presented in the thematic model (See Figure 1). Four themes on the left side of the figure below reflect experiential characteristics of late miscarriage: (1) Initial Splitting state, (2) Betrayal of the Body, (3) Disconnecting, and (4) Reconnecting. The other four themes on the right side of the figure present the practical guidelines on post-natal health care based on the interview data: (5) Informing, (6) Opportunity for Goodbye, (7) Attention to Emotional Wellbeing, and (8) Respectful Hospital Environment. Each of the themes and the related subthemes are described, interpreted, and analysed in detail below, providing illustrative quotations from the study participants. The first letter of a participant’s name is indicated in brackets.

### 3.1. Experiential Characteristics of Late Miscarriage

#### 3.1.1. Initial Splitting State

The participants often defined the primary stage of loss as a splitting state characterised by dissociation, an opened void, and impaired symbiosis between a mother and the foetus, between the old and the emerging identity of a pregnant woman, and between the body and the psyche.

Dissociation. The natural onset of spontaneous abortion and the news of nonviable development of pregnancy were unexpected to the women and were subjectively related to shock reactions. Defensive reactions kicked in, serving as a buffer in helping the women to delay the processing of psychologically traumatic information. On a rational level, participants were able to momentarily process the news of the death of the foetus; however, on an emotional level, they felt as if they were ‘turned off’: ‘<…> I was sort of switched off [laughs] <…> It wasn’t that I was denying the reality. I realised that I had reached a full stop. Well, okay, that it was… That it won’t happen. It’s like some glass or something similar breaks and, I don’t know, some kind of illusion’ (N). The perception of participants’ reality changed during miscarriage. The women were as if in a daze; they made jokes, felt concerned about the feelings of the surrounding people, and performed the doctors’ instructions. Dissociation was further enhanced by the severe pain of ongoing miscarriage: ‘The pain was so intense that I started crawling on the floor [pause]. I had some hallucinations in my eyes’ (Ko). In some cases, dissociation continued even after miscarriage. Such dissociation was characterised by short-term psychotic elements, confusion, and failure to understand what the actual reality was. For example, after miscarriage, Sima was terrified that she might go crazy: ‘I am all alone and, of course, something [clears her throat] terrible has just happened in my life. And I, I am still feeling something that is no longer there, and it seems that I have gone crazy [slowly, in an elevated tone, with a smile].’ The possibility to break down and go crazy seemed to be really close.An opened void. After miscarriage, the women felt as if a physical, psychological, and spiritual void had opened. They defined miscarriage as a significant loss, a disaster, a catastrophe, or even death: ‘that of loss and death… <…> I had that feeling of void. <…> it all seemed so meaningless’ (M). The women experienced a feeling of restlessness and a deep sensation of meaninglessness. In certain cases, the women experienced a physical void in their body, as if an emptiness had opened in their belly.An impaired symbiosis. The women experienced late miscarriage in the stage of symbiosis with the foetus, where they were still unable to distinguish the limits of their body from those of their foetus. Confusion of concepts prevailed when they tried to identify what had happened. The women identified miscarriage with childbirth, used the concepts interchangeably: ‘I remember this after my second childbirth. <…> Well, not childbirth but rather miscarriage’ (Joana). The women regarded early spontaneous abortion as emotionally not so difficult: ‘Say, when you have this easy miscarriage—spontaneous—then you just start bleeding. Here, you have to give birth to someone non-existent [sadly].’ (D). Late miscarriage impaired the symbiosis between a mother and the foetus. In the case of late miscarriage, the pain of loss was further intensified by the already occurred break of the internal prenatal attachment to the foetus often associated with the image of the foetus seen and the heartbeat heard during the ultrasound examination, as well as with the first movements of the foetus. During a longer pregnancy, an internal dialogue with the foetus was taking place—the foetus was personified, referred to as ‘little one, baby, girl, or boy’: ‘We were both <…> convinced to the bone that this time our child would be born’ (S). Symbiosis was being developed not by the duration of pregnancy itself but, rather, by the relationship with the foetus. Some women identified the loss of the foetus with the death of their closest family members: ‘It was like burying the closest person. It doesn’t really matter that like others say, “it is just three months”; that it doesn’t count; that, you know, it wasn’t even nine months and there was nothing there, no relationship. Well, for me, there was the relationship. [sniffles]. <…> And then you have to bury him—in a grave or in your heart’ (M).The body is still pregnant while the psyche is mourning. In this condition of splitting, the dualism of the participant’s body and psyche became apparent. The body and the psyche operated separately, contradicting each other. After miscarriage, certain physical symptoms of continuing pregnancy persisted, e.g., breast enlargement and activation of lactation, nausea, a round pregnant-looking belly, and increasing blood indicators in hormone tests: ‘My breasts were so heavy, milk was being produced; I could feel all that tension and pain, and I kept thinking, “what could I possibly do with that milk?” This… was probably most unpleasant or, how should I put it, most reminiscent of what had just happened <…>’ (J); ‘You are carrying, you feel nauseous. You feel all the symptoms and you understand that it is all meaningless…’ (D). After her miscarriage, Sima could still feel the movements of the foetus: ‘I still had that sensation of the foetus even though it was no longer there. Really. I then started thinking that perhaps I was crazy because I could feel <…> movements.’ (S).

#### 3.1.2. Betrayal of the Body

Following late miscarriage, the women went through a change in their relationship with their bodies. This caused distrust in them. They experienced anger, as if their bodies had betrayed them. The experience of betrayal of the body can be defined by the following subthemes: symbolic experience of internalised death, shocking physicality of ongoing miscarriage, lost control of the body, and confusing body signals. 

Symbolic experience of internalised death. The news that the development of the foetus had stopped and the process of decomposition had begun stirred strong feelings in the women: horror, queasiness, and fear. In some interviews, the image of a walking coffin was employed to define that experience of internalised death: ‘It was very hard. That sensation as if you are carrying an empty child, the feeling that you are like some coffin. Well, in general this somewhat inexplicable feeling. You’re walking…’ (K). With some women, the experience was further aggravated by the process of waiting for spontaneous abortion or planned medical interventions after they had learnt the diagnosis of nonviable pregnancy: ‘I didn’t want to do anything, it was very hard because I had that baby in me that could not possibly develop. And you cannot just get-get rid of it’ (N). That experience of miscarriage was paradoxical, for this constituted both birth and death for the participants: ‘You’re giving birth to your stillborn baby, that little body that had just been fi-finished off with tablets. And you must give birth to it otherwise surgery awaits you’ (Ko). One of the most significant aspects of this situation for the participants was the question as to what should be done with the foetus remains.Shocking materiality of ongoing miscarriage. During miscarriage, the participants were forced to face something that usually remains invisible externally in the case of smooth pregnancy. The women were shocked by the lost integrity of their body. In certain cases, there was excessive bleeding; the women were frightened by the intensity of bleeding, which, in turn, aroused the fear of bleeding to death: ‘I felt very weak, I could hardly stand. My husband was just looking at me and I was barely holding on. So, I told him to call an ambulance because I was feeble and the bleeding was appalling’ (K). In other instances, the women had to deal not with ordinary bleeding but, rather, with shedding of the uterine lining, as well as with other bodily sensations that are hard to describe: ‘I didn’t start to bleed, I began to… well, how can I put it what I felt… Basically, my waters broke. But not in that normal way, but with this, sort of, mucus plug and with something that looked like a balloon…’ (S). During a medically induced miscarriage, the women—in a conscious state—had to give birth to their dead foetus. This was a continuous, painful, complicated, and a rather long process: ‘They said, “it got stuck. Well, the head came out, but, well, the legs got stuck. So, we’ll give you one tablet and we’ll wait for it to come out.” And they gave me that tablet. Dusk fell and I was just lying there waiting for it to take effect. [long pause]’ (Ko).Lost control of the body. Helplessness of the participants was further enhanced by the fact that their bodies had disobeyed them. Both the women and their foetuses were described as hostages of the body: ‘She said that there weren’t any more waters left but the baby’s heart was still beating. But we couldn’t just produce or somehow inject them. So, she said that we had to wait and see what would happen. And, of course, it was evident what would happen… We just stayed in hospital to observe, and the following morning the heartbeat was gone’ (J). The body became a kind of scapegoat—it was to be blamed for miscarriage; it was regarded as inadequate, defective or damaged in contrast to the bodies of other women who gave birth successfully. In some cases, new pregnancy helped to justify the undervalued body: ‘The relief that I got pregnant <…> I was, sort of, glad that this theory or fantasy—you know <…> that it tends to repeat—hadn’t been confirmed; that I can somehow or that I am capable of, and my body isn’t totally helpless [laughs]’ (N).Confusing body signals. The physical symptoms of pregnancy pathology before miscarriage were not clear enough. Some women were able to identify them only in retrospect, after the loss: ‘Now that I try to analyse, I felt like I had this, like some kind of [laughs] a foreign body in me, terrible as it may sound. Well, because I was constantly like… as if I had food poisoning. Those four months, three months were like never-ending food poisoning’ (S). In some situations, the body did not send any signals at all, and pregnancy pathology was betrayed only by inexplicable intuition: ‘From the very first weeks I felt a strong—I don’t know—some really intense anxiety. I mean the premonition was really powerful’ (Ko). In other cases, the body symptoms contradicted the fact of nonviable pregnancy, which only intensified the women’s anger towards their bodies; they felt deceived by false body signals: ‘This body of mine, well, it didn’t send any signals; well, it was like <...> such a betrayal on my body’s part <…>.’ (N).

#### 3.1.3. Disconnecting

Miscarriage shook the participants’ foundations of safety. This was the experience that isolated the women from the outside world. The women found themselves in a depersonalising hospital environment, while the guilt for miscarrying distorted their perceptions, and in the end, they took all responsibility for the reasons and solutions of the miscarriage solely on themselves. Retreat was the only way to protect oneself and not to break down during the initial stage of loss.

Depersonalising hospital environment. A medicalised approach of the health care staff that did not meet the emotional and spiritual needs associated with prenatal loss was frequently mentioned in most of the stories. The women were affected by such underestimation of loss, when only their body was being treated in hospital: ‘<…> felt “patientalised,” if I could say so. I was just an ordinary case. You know, someone was going to be operated on appendicitis, I was supposed to push the child out, yet somebody else had to have their burnt leg bandaged. Oh, of course, the sterility differed, but in essence, everything was the same. <…> your leg will heal, you haven’t lost anything, <…> but this is still a part of your body, not some other person in you’ (Ko). While hospitalised, the women often felt depersonalised and treated merely as statistical units. The women experienced macabre situations when they were given an empty mayonnaise bottle or a bucket for bleeding or had to wait for miscarriage in a toilet. The need for privacy was not satisfied. In some cases, the miscarriage took place in a general ward, with strangers present. Loss was not recognised in the hospital environment, and mourning was suppressed and hushed; the women felt under pressure to mourn in solitude. The cold and emotionally distant reactions of medical staff during the performed interventions stood out in great contrast to the intensity of emotions experienced by the women at that time. Loss was realistic only for the women but not for the external environment. The women had to cope with the feeling of existential loneliness: ‘I was alone [sniffles], there was no one I could, I mean, nobody supported me [with a breaking voice, cries out loud, a long pause, sniffles, a long pause]. I was sitting in the corridor [in hospital] crying’ (M). The participants also lacked humanness, care, and emotional support from health care personnel.Guilt falsifies perception. An interpretative analysis revealed that the guilt for miscarriage distorted the participants’ ability to assess the situation. They mainly attributed the responsibility for the reasons of miscarriage to themselves. The women blamed themselves for the decision regarding the method of medical intervention (medically induced miscarriage or dilatation and curettage): ‘This was my decision [sniffles]. And when I was admitted to hospital, I said that I would be murdering the baby with my own hands. So, that moment was morally and otherwise the most painful’ (M). The participants blamed themselves, saying that, in that hospital environment, they did not fight for the possibility to bury their dead foetuses. They also assumed responsibility for the reactions of the surrounding people (partners and parents) to the miscarriage: ‘You feel guilty that others are also suffering’ (D). The guilt experienced in the state of crisis vulnerability distorted the way the women interpreted reality, and often, the resulting aggression was directed against the women themselves. Some of the participants even considered the possibility of committing suicide.Retreat as a means of self-preservation. After miscarriage, the women experienced the need to isolate themselves from the external environment by setting strict boundaries of privacy. The need for safety at the beginning was much greater than the need for social support. In their special state of vulnerability, the women chose to hide in a cave and lick their wounds. At first, they were exceedingly reluctant to share their miscarriage experiences and sought to avoid any meetings with friends, fearing that the people around them might not understand them, dreading inadequate and unemphatic comments. In the future, when the women got pregnant again, they tended to hide the news about their pregnancy from the people surrounding them for a long period of time. Retreat was necessary for the women to integrate their experiences into their identity, to discover their own answers, to accept their loss, and to gradually come back to their social lives. In the initial stage, such a retreat was their way to survive and not to break down. Exposure to their vulnerability increased immensely when they encountered other pregnant women or those who had managed to give birth successfully: ‘Right above us, in our block of flats, there was this woman, our neighbour who had given birth two months before. Her baby would cry at nights, and I couldn’t sleep. I heard everything and I had that feeling that it was my child crying above my head, in heavens’ (K).

#### 3.1.4. Reconnecting

Eventually, faced with the need to integrate the experience of miscarriage into their life stories, the women started looking for ways how to restore their foundations of safety and, gradually, to come back to their social lives. This topic defines the stage of miscarriage acceptance and gives meaning to it, which can be further revealed in the following two subthemes: collecting shatters, as well as reinterpretation of pregnancy and the phenomena of maternity.

Collecting shatters. Physical recovery after miscarriage occurred much faster than emotional and spiritual acceptance. Neringa discovered that the denial of loss experience was harmful to her: ‘Then somehow you have to collect those shatters. Well, somehow it really seems to me that it is so harmful to me; well, if I don’t work with that [laughs]’ (N). In some instances, avoidance of the feelings of loss may be associated with complicated and prolonged mourning: ‘Such things don’t just go away by themselves. It’s like in a dream where I bury everything in a large hole… It’s not a solution. Later, that rotting and that smell come from within; oh no, there’s nothing good out of it’ (M). Symbolic, respectful farewell to their foetus, acceptance of the feelings of loss, and reflections on them in a safe environment, individual psychotherapy and the support of their closest people—especially a woman helping another woman—helped the women ‘collect shatters’. Even though the significance of their miscarriages remained fairly intensive until the study, most of them succeeded in accepting it. The women tended to identify the experience with a scar. Despite the fact that pain subsides over time, miscarriage marks persist, because such an experience is indelible.Reinterpretation of maternal identity. Miscarriage leaves an imprint on future pregnancies. It changed the women’s approach to pregnancy—from pregnancy as a natural process (‘pregnancy is not a disease’) to pregnancy as a state requiring tranquillity and concentration: ‘I would say to myself, “Come what may. You can’t just sit here with your legs crossed. Well, you just can’t stop it.” <…> And then, when we experienced our first miscarriage, I started thinking about this a lot; you know, that perhaps I should have been more careful, I should have been more… afraid, as they say—I should have been like that “mother hen.” [laughs]’ (J). With their subsequent pregnancies, the women felt as if that joy of pregnancy was taken away from them. An anxious state of guarding and vigilance prevailed. The women tried to control everything by observing compulsively every symptom of their body. Some women kept checking on a frequent basis whether there was no bloody discharge: ‘I would check myself every time I left the toilet, every single time. So, I don’t know how many thousands of times there were in total’ (Ko). The women and their partners did their best not to develop a strong attachment to the new pregnancy and were ready for the worst. In some cases, the subjective psychological effects of miscarriage also reverberated in maternity—the participants were inclined to evaluate themselves as anxious mothers. Nevertheless, it is this experience of maternity that enabled most of the participants to reintegrate their miscarriage experiences into their identity as women, as well as to restore their foundations of safety: ‘Perhaps that general feeling, not that I am a mother, but rather the fact that I am able, that I was able to walk this entire path <…> It’s a really significant moment for a woman. That I am absolutely, well, totally capable of doing it. That feeling of restoration of self-confidence.’ (N).

### 3.2. Practical Implications in Post-Natal Health Care

Theme’s “Practical Implications for Post-Natal Health Care” Subthemes and Categories are represented in the table (see Table 2).

#### 3.2.1. Informing

Not to overwhelm women with excessive information when they are in a state of shock. Confusion and shock constitute a natural reaction to the news of miscarriage or nonviable pregnancy. The experience is more of emotional rather than rational nature. During those first moments, it can be really difficult for patients to understand the information provided by a doctor: ‘But at first they told me in hospital <…> “Do you have any questions?” But when this happens, you just don’t know what you could possibly ask’ (J). It is important to accept a patient’s confusion and give her some time to comprehend the news. The information provided by medical staff will be perceived more efficiently after some time, when the initial spontaneous emotions subside and a woman gets out of that ‘red’ zone of fight–flight–freeze. In the case of loss, rationalisation and depersonalised informing are not the right means of providing first aid. Answers to questions of concern become important a bit later. Therefore, it is crucial to ensure that a patient is able to get in touch with her doctor in the future, for example, by indicating a telephone number or visiting the hospitalised patient later to repeat the information.To ensure a safe and private environment to accept the news. During those first moments after having just learnt of ongoing miscarriage or nonviable pregnancy, women are in much greater need of emotional support, expression of care, and consideration rather than information: ‘Well, there is, there is this stress, anyway. You know, that he is dead. You’re already speechless and, well, uhh, you don’t have any questions. You need help at that time’ (K). Ensuring safety and comfort is actually something that heals. Publicity increases insecurity, since there is a greater risk of experiencing exposure to the incurred psychological trauma when seeing other healthy pregnant women or women who have recently given birth to healthy babies in a hospital environment.Not to provide secondary information. Giving secondary information related to future pregnancies, cancer prevention programmes, etc. is not recommended immediately after delivering the news of stopped foetal development, unless a patient asks about this and needs advice: ‘[The doctor] says, “Recommendations regarding conception—next time.” And I’m just thinking to myself that right now I’m not interested in any conception at all’ (J).To explain physiological processes in a simple and understandable manner. Having processed the initial information, women need a thorough explanation of the ongoing physiological processes given in a simple language. It is highly important to provide a clear and systematic description of a further course of treatment: ‘A woman does need clear steps as to what has to be done with this now, and the doctor, “Oh, it will all pass [by itself]” [with irony]’ (K). Clarity and precision offer much-needed safety and help to restore the lost feeling of control.To avoid inaccurate considerations. Uncertainty is further intensified by inaccurate considerations of their emotional nature: ‘He said, “Oh, it’s like winning a million in a lottery.” Yeah, lucky, but unlucky. And I just thought to myself, “How can you possibly compare this to some lottery” [in a raised voice, laughs], <…> So, I said that maybe it was high time to start looking at what could be done specifically, that maybe we should do tests instead of playing a lottery… I remember I didn’t like it at all and I never went to him again’ (J). Meaningless talk and philosophising considerations are of no use in a state of shock. A human being is incapable of abstract reflections and does not understand the figurative sense due to the deep impact of the situation. Black humour or irony are also incomprehensible. Therefore, it is highly important to select words carefully, to combine human empathy with scientific, research-based information.To dedicate sufficient time. A constant rush of medical staff causes tension, and a patient can feel unheard. Disappointment in provided services can be further enhanced by a doctor’s delay and a long waiting time in the queue: ‘[imitating her doctor] “I don’t have time now. Here, do this, this, this test. See you.” Oh no. I’m not coming back to this [doctor], ever’ (J). It is difficult to process information due to the constant haste of doctors: ‘I lacked information. Meh… from doctors. [long pause] But some simple details. At that time, we, perhaps then we, out of all this… eh… this hectic pace’ (S). On the contrary, satisfaction in provided health care services increases immensely when sufficient time is dedicated to informing: ‘That shift was just fantastic. And that obstetrician, who was on duty, she really devoted more time to me than, I believe, they can or have in those state hospitals [emphasizes with irony]’ (S).

#### 3.2.2. Opportunity for Goodbye

To provide an opportunity to bid farewell to a dead foetus. The women really appreciated the possibility to say goodbye to their dead foetus. Some participants indicated that it was really important for them to see or touch the embryo/foetus, to get the foetal remains back and to have the possibility to bury them: ‘During the second time, they asked me if I wanted to see the baby. When it first happened, they just sort of took it, covered it and brought it out. <…> I don’t remember whether they offered us to take a look. And that second time, they even made those footprints on a postcard with paint; they put it in a box (they have special boxes) and there were flowers brought to doctors by other pregnant women, so they put flowers, cut flower blossoms into that box, and put that baby in it’ (J). The meeting with the foetal remains helped the women realise their loss and reduced their state of splitting. In the short term, this can provoke intensive feelings of loss; however, in the long term, this enables a smoother process of mourning, diminishes the subjective risk of denial, ignoring such feelings or pushing them away, thus complicating the mourning. A real tactile and visual encounter with loss enables women to subsequently integrate it into their experience and reduce the feeling of guilt: ‘I had already had this information when someone had experienced miscarriage and wanted to bury the child, and they had looked for ways how to do it. Because for me, this is life. It’s not just a mere organ that needs to be removed, thrown out’ (K). It is important to ensure sufficient time to say goodbye to foetal remains in a private environment: ‘They brought that box with the baby, and they brought a candle and told me, “Stay with the baby. Bid your farewells. Take your time, as long as you need.”’ (J).To offer the possibility to choose how to say goodbye to foetal remains. It is important to offer clear, equivalent alternatives as to what can be done with the foetal remains. There is no single correct or incorrect way; however, it is highly important to ensure that a woman has the possibility to choose the most morally acceptable alternative: ‘They let us do what we wanted to. Whether we wanted to bury, or to cremate, or just to say that we didn’t want, we didn’t want to see or to do anything with it’ (J). It is absolutely normal that such a moral decision can change in a state of shock: ‘I had prepared everything. I just wanted to put him into a box and take him with me and instead I… I just said that I didn’t, didn’t want them to give him to me’ (M). In any case, it is necessary to respect the choice of a woman and her partner. The most important thing is not to disempower a woman and not to strip her of her right to make her own decision regarding the best way to say goodbye to the foetal remains: ‘Actually, I wanted to see her, but nobody showed her to me. They said that it was better not to. I don’t even know whether I actually said so, you know, that I wanted to see her that very moment, but there was this feeling that I did. And… and they just brought that mayonnaise bottle out…’ (Ko). Disempowerment strengthens the feeling of inferiority.

#### 3.2.3. Attention to Emotional Wellbeing

To offer initial emotional support. Emotional support from the doctor was one of the most significant aspects for the patients. A doctor’s humanness, accessibility, and expressed concern help a patient to accept the feelings of loss. In situations like this, the patients felt understood, empowered, and placed more trust in the health care system: ‘Simply, in simple words, directly admitting that those were the most painful situations at work for her as well; and she did show interest in me asking about my previous experiences, well, simply caring about me. I do remember such episodes from our conversations. Very simple language, really, but… I felt cared for’ (S). Acceptance and validation of the feelings of loss were very important: ‘Her [doctor’s] reaction to my tears was so sensitive’ (Ko). Sometimes, a simple wordless expression of compassion served as significant support to the women: ‘Well, I remained there and, after that cleaning, the doctor came. She asked me how I was feeling and took my hand. And then I grabbed her hand and said, “How good of you to have taken my hand. Well, I just needed it so badly…”’ (K).To ensure that health care professionals are adequately prepared to provide initial emotional support. Health care professionals need to be trained to provide initial emotional support. Insecurity is only strengthened by a doctor’s confusion and inability to react properly in emotional situations: ‘I think the main doctor didn’t say anything at all. And later, this other doctor said that… but he was very, how to put it…modest or something of the kind. It looked as if he had been preparing all day long to tell me that everything would be alright’ (Ko). Such comments as ‘everything will be okay’ or ‘next time you’ll return to hospital to give birth’ are not suitable in the case of loss.To evaluate emotional wellbeing during a repeat visit to the gynaecologist. During a repeat inspection, women should be asked open questions about their emotional wellbeing, which would help differentiate the need for psychological help: ‘Perhaps after some six weeks, when you have to go to the doctor. Then, one way or another, maybe by asking how we were feeling, they would see whether there was some depression or that perhaps something was coming; that you just couldn’t go through this, that you were still very distressed, very upset. Then maybe they would direct you to someone else’ (J).To refer patients to professional psychological help. The results have revealed that the availability of psychological help was usually not ensured: ‘I asked about that help. I asked where I could get professional help. Well, you had just given birth to your dead, ummm, child. I don’t know. So, they said there was none’ (Ko). The doctors were not ready to inform the patients about the possibility to be referred to psychological help: ‘So, the doctor comes and asks, “How’re you feeling?” And I say, “Bad.” “What happened? What is it? Does it hurt anywhere?” “No, it doesn’t hurt, but I’m feeling bad psychologically <…> I need psychological help.” And the doctor replies, “Well, for your information, we do not offer such help.”’ (Kristina). In medical institutions, psychological help should be proactive, i.e., when a patient is clearly referred to a specific psychologist or a psychologist initiates that primary contact and provides the information regarding the possibility of consultation. An offer of assistance alone may have a therapeutical effect, since this helps a woman to accept the fact that it is absolutely normal to experience a feeling of loss and that such need for help is totally normal.To engage a partner. At the time of loss, the closest people constituted the most significant source of emotional support. Something that feels safe and familiar helps most in a state of vulnerability. Going through a miscarriage together with a partner and their emotional engagement in the process serves as considerable factors of help. Health care personnel should allow a partner to participate in gynaecologist consultations when presenting the diagnosis or the results of other important tests. It is crucial to promote a shared responsibility and joint decision-making process of a couple when making decisions concerning interventions and to allow a partner to stay with the patient in a hospital environment while waiting for interventions after a miscarriage: ‘I would have liked my husband to be with me, at least for a night. Well, that he could stay for a night and, of course, I wanted that badly at that time. I wished that that he could stay and those would have been the perfect conditions for me to get through this…’ (S).

#### 3.2.4. Respectful Hospital Environment

The importance of an aesthetic environment. The hospital environment scared some of the study participants, creating a macabre feeling and sense of depersonalisation. In some instances, during miscarriage, the women were given an empty mayonnaise bottle or a plastic bed pan or were instructed to wait out the contractions and the urge to push in a toilet. A cold, uncomfortable, harsh, unpleasant atmosphere can intensify the feelings of loss, as well as the experienced terror. On the contrary, a cosy and aesthetic ward can contribute partly to a better emotional wellbeing.The distribution of wards. During miscarriage or when waiting for miscarriage, an obvious need for privacy was observed. The presence of strangers had a negative effect on the participants: ‘I was somewhat shocked by that, by that… of the ward <…> when I was going through my worst pain, the entire family was present… Uhh… visiting that woman. So, I asked and even my husband asked them to leave because I was feeling really bad’ (Ko). The women were obliged to suppress their natural feelings of loss because of the presence of other patients, even though emotional discharge would have brought relief: ‘I was brought back to the ward and I couldn’t sleep, I was just sobbing. And there was this other woman, and I was worried that I might wake her up because the pain and grief were profound’ (K). The experience of depersonalised hospitalisation along with other patients was hurtful. Hospitalisation in the same ward with successfully pregnant women, women after abortions, or women who had given birth to healthy children served as a repeated traumatic experience.Respect for the body. It is highly important to ensure respect of the body of a woman and that of her foetus. In the case of late miscarriage, the women felt exhausted from the pain of contractions and the urge to push. Dalia was not offered any painkillers, and interventions were performed in a rough way: ‘I was pushing that placenta, but since they had mixed everything inside me before that, I guess, something, some bits and part must have remained. And they couldn’t just leave them there for this could cause severe bleeding. So, we went to another room, and they tried to scrape everything out with those metal instruments’ (Ko).

## 4. Discussion

A qualitative analysis revealed a four-stage process during which the study participants experienced late miscarriage. The first stage was described as an initial splitting state, which was characterised by dissociation, emptiness, an impaired symbiosis between women and foetus, and dualism between women’s body and psyche. These experiences are likely to unfold in relation to broken prenatal attachment—a unique relationship between mother and foetus [36]. This state of disintegration is likely to be more characteristic of miscarriage in the second trimester of pregnancy, as, according to Donald Winnicott (1956), the woman had already entered ‘a very special state of the mother, a psychological condition which deserves a name, such as Primary Maternal Preoccupation’ [37]. The second stage of the inner process after late miscarriage reveals an embodied late miscarriage experience known as a betrayal of the body, which was characterised by symbolic internalised death experience, shocking physicality, lost control, and confusing body signals. Qualitative research in the literature highlights the central role of the lived body experience in pregnancy loss bereavement [38]. The third stage is the state of disconnection from the outside world, which was experienced by depersonalising medical environment, guilt, and withdrawal as a means of self-preservation. Finally, the last stage in this process is reconnecting, which was described as the necessity to collect shatters and acquired a reinterpretation of maternal identity. The significance of miscarriage in motherhood was associated with existential transformation and the experience of the fragility of life. This was also reflected in a comprehensive literature review, which showed that the risk of a more intense grief and distress reactions correlated with a strong previous desire for the pregnancy, difficulties getting pregnant, having no other children, experiencing other losses in the past, later gestational week, having little social support, and poor coping strategies [22].

Practical implications for post-natal health care include clear and attentive informing, a provided opportunity for saying goodbye, sufficient attention to emotional wellbeing, and a respectful hospital environment. In this study, Lithuanian participants who experienced a miscarriage named the need for specialised psychological help. The literature showed that predictors of the development of complicated grief after prenatal loss include a lack of social support, as well as ambivalent attitudes or heightened perception of the reality of the pregnancy [25]. Accordingly, it is emphasised that, given the range of potential meanings for this primarily prospective and symbolic loss, practitioners need to encourage patients to articulate the specific nature of their loss and assist in helping them cement the experience [22]. One of the key questions is how to help partners become fully involved, as they are among the most important sources of support for women. As the research shows, partners are also affected by miscarriage experiences and might need professional support themselves [27,28]. Family-centred social support could be relevant. Furthermore, the data suggest that greater targeted support and monitoring for women who have a history of mental health problems may assist those women in coping following miscarriage [39].

There is a contradiction between the findings of quantitative and qualitative research on the topic of miscarriage. Quantitative research shows that women who miscarry usually experience moderate depression and anxiety, which persists for around 6 months, though the qualitative analysis of the interviews indicates that women who have had miscarriages experience deep emotional responses and a longer and more difficult process of coming to terms with their loss [40,41]. This may be related to the tendency that more vulnerable women show interest in participating in voluntary qualitative research. However, it is the needs of these women that are crucial to respond to while providing health care and emotional support. It can also be assumed that some aspects of the miscarriage experience are too in-depth and intrinsic and therefore difficult to capture in quantitative studies.

A miscarriage experience is an important public health issue. A qualitative research strategy was applied in order to develop a coherent theoretical construct for survival after miscarriage and coping with grief after prenatal loss. The experiential analysis revealed specific stages in the process of accepting women’s miscarriages and reconciling. In this study, particular attention was paid to the prevention of women’s complicated grief and the aspects of successful adjustment after a miscarriage. The physical, psychological, and social health aspects are equally covered. A comprehensive in-depth analysis not only helped to reveal women’s experiences in detail but also yielded information of targeted preventive measures and practical recommendations for post-natal health care based on women’s experiences. The major limitation of this study is the small study sample, and the conclusions are limited for this reason. Though a small study sample was the basis for a thorough analysis of individual experiences, larger-scale representative studies would allow the validity of these qualitative analysis findings to be verified. Several other important limitations of the study are that the results of the qualitative study cannot be generalised, and the original theoretical model should be verified by quantitative studies involving a representative sample of the participants. Whereas qualitative research does not aim to compare groups, there was no comparative analysis in connection to health conditions, psychiatric conditions, substance and medication use, etc. These aspects could be relevant for future quantitative or mixed-method studies. Moreover, in future research, it is worth exploring other specific situations related to miscarriages: the experience of recurrent miscarriages, men’s post-miscarriage experiences, and others.

## 5. Conclusions

The experiential characteristics of late miscarriage were described by four themes and 13 subthemes: the initial splitting state, Betrayal of the body, Disconnecting, and Reconnecting.After a late miscarriage, it is important to prioritise the following needs of women: creating a safe environment for receiving sensitive information, not denying their feelings of mourning, providing not only medical but also emotional support, and encouraging partner involvement.Practical implications for post-natal health care include clear and attentive informing, provided the opportunity for saying goodbye, sufficient attention to their emotional wellbeing, and a respectful hospital environment. Lithuanian participants who experienced a miscarriage named the need for specialised psychological help.

## Figures and Tables

**Figure 1 healthcare-10-00079-f001:**
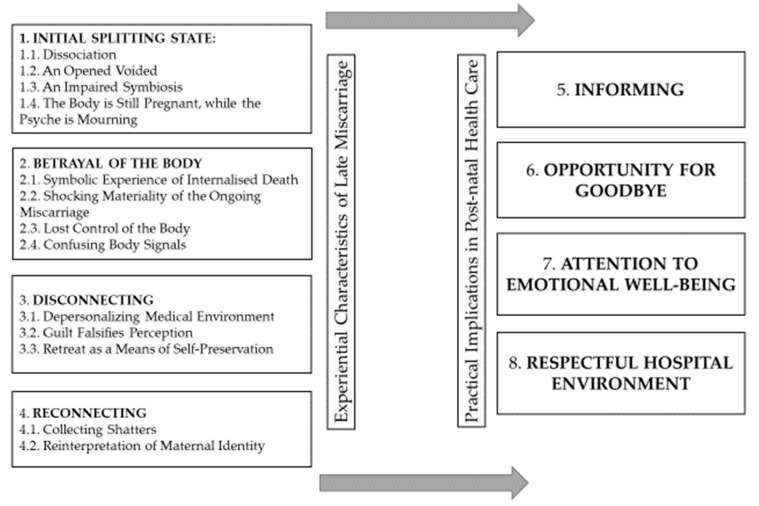
Thematic model: experiential characteristics of late miscarriage and practical implications in post-natal health care.

**Table 1 healthcare-10-00079-t001:** Demographic characteristics of a study sample.

Anonymised Name	Age	Number of Miscarriages (Year)	Gestational Week	Number of Children in the Family (Age)
Joana	33 y	2 (2014; 2015)	20 w; 19 w	1 (2.5 y)
Sima	31 y	2 (2015; 2016)	6 w; 17 w	1 (2 y)
Kotryna	33 y	2 (2016; 2017)	8 w; 17 w	2 (4 y; 1 y)
Miglė	40 y	2 (2016; 2018)	12 w; 19–20 w	1 (6.5 y)
Neringa	38 y	1 (2015)	18 w	1 (4 y)
Kristina	52 y	3 (2009; 2012; 2015)	20 w; 5 w; 10 w	5 (25 y; 16 y; 13 y; 10 y; 8 y)
Danguolė	29 y	1 (2020)	14 w	-

**Table 2 healthcare-10-00079-t002:** Practical implications for post-natal health care.

Theme “Practical Implications for Post-Natal Health Care” Subthemes	Categories on Practical Implications
Informing	Not to overwhelm women with excessive information when they are in a state of shock
	2. To ensure a safe and private environment to accept the news
	3. Not to provide secondary information
	4. To explain physiological processes in a simple and understandable manner
	5. To avoid inaccurate considerations
	6. To dedicate sufficient time
Opportunity For Goodbye	7. To provide an opportunity to bid farewell to a dead foetus
	8. To offer the possibility to choose how to say goodbye to foetal remains
Attention To Emotional Wellbeing	9. To offer initial emotional support
	10. To ensure that health care professionals are adequately prepared to provide initial emotional support
	11. To evaluate emotional wellbeing during a repeat visit to the gynaecologist
	12. To refer patients to professional psychological help
	13. To engage a partner
Respectful Hospital Environment	14. The importance of an aesthetic environment
	15. The distribution of wards
	16. Respect for the body

## Data Availability

Anonymised databases are available from the corresponding author upon reasonable request.

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
