# Peer review of "Experience of Late Miscarriage and Practical Implications for Post-Natal Health Care: Qualitative Study"

_healthcare, 2022, doi:10.3390/healthcare10010079_

Round 1

Reviewer 1 Report

Interesting study, seems to be a bit of a pilot consider the size, conclusions from the study are limited for this reason.

Introduction

  • Move all references to the end of the sentence
  • Add US reference as to the definition of pregnancy loss/miscarriage
  • Paragraph 2 seems to spend time on the female perspective, while there are many male partners who may be affected by the loss. Recommend expanding on this angle.
  • Additionally, recommend removing binary terms wherever possible, using the term persons assigned female at birth, etc. Line 52- this is really a ‘reproductive health’ issue not a women’s issue.

Materials and Methods

  • 1: How were women recruited/selected? What was the criteria for inclusion? Did all women have a hospital assisted miscarriage? Did the interviewer know or have previous relationship with the women? This may have impacted the ability of the patient to open up about experiences.
  • What is meant by “abrasion”? line 80

Results

  • For the section 3.1 following Figure 1, several statements of fact “eg. This caused this” should be softened to this may have impacted, this may have lead to… etc. Such definitive statements not usually made outside of an RCT, as cause and effect cannot be determined by a 7 patient observational study
  • Please shorten 3.1, maybe to a few examples for each section
  • Thematic elements should provide data for how many of the patients responded under each of the themes. Example – did all women have a hospital based miscarriage? This is not the standard in other countries and if it is in Lithuania, the miscarriage process should be described in more detail in the introduction.

Discussion

  • What type of health care - national or privatized and does this have an impact on the patients?
  • The small sample size should be discussed as a major limitation
  • The male factor/partner factor as identified above should be discussed
  • Were all females heterosexual, married, etc? More participant demographics would be useful – other chronic health conditions, other psychiatric conditions, substance/alcohol use, medication use, BMI, more.

Conclusions

  • Recommend remove numbers from the conclusion and eliminate #1 as redundant from results. What are future steps, what is the plan and what to do with this information should be expanded upon.

Author Response

I am sincerely thankful for your valuable insights and an in-depth review that will definitely help improve this article. I have taken your observations carefully into account when making corrections to the article. In those points where corrections couldn’t have been made, I will try to argue and substantiate methodological decisions in more detail. Please see the attachment file.

Response to Reviewer 1 Comments

I am sincerely thankful for your valuable insights and an in-depth review that will definitely help improve this article. I have taken your observations carefully into account when making corrections to the article. In those points where corrections couldn’t have been made, I will try to argue and substantiate methodological decisions in more detail.

Point 1: Interesting study, seems to be a bit of a pilot consider the size, conclusions from the study are limited for this reason.

Response 1: Thank you for your interest in this study. In this qualitative study phenomenological paradigm and Thematic analysis method had been chosen. According to V. Braun and V. Clarke, who are one of the most cited Thematic analysis methodology experts, for experiential studies small samples are suitable. Decision on the number of participants (n-7) was based on “data saturation”: sample had to be large enough “to demonstrate patterns across a date set, but small enough to retain a focus on the experiences of individual participant” (Braun & Clarke, 2013, p. 45). According to J. A. Smith and M. Osbourn, who represent phenomenological paradigm, phenomenological studies have been published with samples of one, four, nine or fifteen participants, and for the beginners they suggest three as a perfect sample, which allows “sufficient in-depth engagement with each individual case but also allows a detailed examination of similarity and difference, convergence and divergence.”; These authors state, that then the sample size is too large, the researcher might “become overwhelmed by the vast amount of data generated by a qualitative study and are not able to produce a sufficiently penetrating analysis” (Smith & Osbourn, 2007, p. 56-57).

The interviews in this study had a leading biographical question and each lasted around 1 hour. Rich sets of complex and sensitive data were generated from each participant. As a relatively rare phenomenon is being studied and late miscarriages account for a very small percentage of all prenatal losses, it was quite difficult to find such participants. Given the sensitivity of the topic and the depth of data collected, it was decided that the number of participants was sufficient for this study. In this study we didn’t aim to provide generalized conclusions, but we wanted to generate inductive descriptions and localized explanations, grounded in unique participant’s experience. Insights from this study should be tested in future quantitative studies.

Point 2: Introduction. Move all references to the end of the sentence. Add US reference as to the definition of pregnancy loss/miscarriage. Paragraph 2 seems to spend time on the female perspective, while there are many male partners who may be affected by the loss. Recommend expanding on this angle. Additionally, recommend removing binary terms wherever possible, using the term persons assigned female at birth, etc. Line 52- this is really a ‘reproductive health’ issue not a women’s issue.

Response 2: Corrections in the Introduction section were made based on these comments.

Point 3: How were women recruited/selected?

Response 3: Corrections in the Methodological section were made based on this question. As the qualitative research doesn’t aim to provide generalized results, but rather seek to provide an in-depth information on specific cases, the typical purposive sampling strategy had been used. Participants had been selected on the basis that they will be able to provide information-rich data to analyse (Braun & Clarke, 2013, p. 58). We applied Snow-ball sampling and searched for Lithuanian women, who experienced miscarriage, in various Facebook and Instagram social networks pages, blogs and groups, e.g. “KÅ«dikio netektis“ (Infant loss), “Krizinio nÄ—štumo centras“ (Critical pregnancy centre), “Viskas apie nÄ—štumÄ…“ (All about the pregnancy), etc. Several public figures, specializing in reproductive health, medicine or psychology were asked to share the invitation to the interviews.  

Point 4: What was the criteria for inclusion? Did all women have a hospital assisted miscarriage?

Response 4:

Inclusion criteria were:

  • Miscarriage happened more than 3 months ago;
  • Gender of possible participant – female;
  • The onset of miscarriage was spontaneous or an undeveloped pregnancy was diagnosed at a gynaecologist’s visit;
  • The participant voluntarily expressed a desire to participate in the interview, reacting to the public invitation advertisement (additional attention is paid to research ethics, considering the sensitivity of the topic and the vulnerable group).

17 other interviews were performed with women, who had had early miscarriages, and 7 interviews – with women who had late miscarriages. The latter sample is analysed in this publication, analysis on early miscarriages is still ongoing and results hadn’t been publicized. All these 7 women had one or more hospital assisted late miscarriage and needed medical intervention, because of the specifics and complexity of late miscarriage.

Point 5: Did the interviewer know or have previous relationship with the women? This may have impacted the ability of the patient to open up about experiences.

Response 5: The interviewer didn’t personally know or didn’t have previous relationship with any of the participants. We had a one-time meeting.

Point 6: What is meant by “abrasion”? line 80

Response 6: Please excuse the inaccurate use of the term due to the peculiarities of the translation. The meaning of this term was surgical intervention. This aspect has been corrected in the publication.

Point 7: For the section 3.1 following Figure 1, several statements of fact “eg. This caused this” should be softened to this may have impacted, this may have lead to… etc. Such definitive statements not usually made outside of an RCT, as cause and effect cannot be determined by a 7-patient observational study.

Please shorten 3.1, maybe to a few examples for each section.

Response 7: Thank you for this remark. Corrections in the 3.1. section were made based on these comments.

Point 8: Thematic elements should provide data for how many of the patients responded under each of the themes. Example – did all women have a hospital-based miscarriage? This is not the standard in other countries and if it is in Lithuania, the miscarriage process should be described in more detail in the introduction.

Response 8: Description of the sample characteristics and aspects of miscarriage process, experienced by the participants, was elaborated in the Methodological part.

Qualitative paradigm, according to V. Braun and V. Clarke, uses the qualitative data, and is the analysis of words which are not reducible to numbers (Braun & Clarke, 2013, p. 4). Differently than in content or other type deductive analysis, in this inductive thematic analysis study we were interested in the experiential analysis of thematic characteristics and meanings. For each sub-theme to be included in the results it had to be mentioned by more than a half of the participants (4 or more). All of the main themes were revealed by all of the 7 participants. In accordance with other inductive qualitative analysis publications, we decided not to include data of frequencies near each thematic element, because the focus was not on frequency but the analytical aspects of semantic and latent content. For instance, in qualitative analysis the most sensitive topics may have a small number of repetitions or be mentioned only by several participants, but may be the most relevant and important.

Point 9:  What type of health care - national or privatized and does this have an impact on the patients? The small sample size should be discussed as a major limitation. The male factor/partner factor as identified above should be discussed. Were all females heterosexual, married, etc? More participant demographics would be useful – other chronic health conditions, other psychiatric conditions, substance/alcohol use, medication use, BMI, more.

Response 9: In Lithuania medical help after late miscarriages is provided only in tertiary or secondary health care sectors, which mostly are national. All the participants received national type of health care. This aspect was elaborated in the methodology part. The sample size was discussed in the discussion part as a major limitation. Male factor/partner factor aspect was also elaborated in discussion part. All of the participants were heterosexual, married or living with their partners, which are the specifics of the sample and might surely affect the conclusions. This aspect was additionally discussed. Whereas qualitative research does not aim to compare groups, there was no comparative analysis in connection to health conditions, psychiatric conditions, substance and medication use. This aspect would be relevant for future quantitative studies.

Point 10: Recommend remove numbers from the conclusion and eliminate #1 as redundant from results. What are future steps, what is the plan and what to do with this information should be expanded upon.

Response 10: Corrections in the Conclusions section were made. Insights for future research were expanded in the Discussion section.

Reviewer 2 Report

The study is very interesting and provides much-needed information on miscarriages

The theoretical review is complete and sufficient to justify the study carried out.

The method is clearly justified and is a suitable methodology for the purpose of the investigation. only, on Line 99 is missing the acronym for Thematic Analysis (TA). In the rest of the paragraph use the acronym.

Results: The authors make an adequate and complete explanation of the results obtained, adequately justifying the narrative they include. Table 2 is very relevant because it summarizes very well the practical implications of the study.

The study finds something very interesting about the stages that women go through, which are not detected in quantitative cross-sectional investigations, which is why the study is very relevant and necessary

The discussion and conclusions of the study conform to the data found and do not make inappropriate generalizations.

For all the above, in my opinion the article is very relevant and necessary and deserves to be published.

Author Response

I am sincerely grateful for Your review, especially the observation that the phenomenon and qualitative approach of the study are relevant, and the study provides interesting and necessary results on the stages women go through after miscarriage. Thank you for delving into this article and providing valuable and encouraging insights from a qualitative research perspective.

Response to Reviewer 2 Comments

Point 1: The study is very interesting and provides much-needed information on miscarriages. The theoretical review is complete and sufficient to justify the study carried out.

Results: The authors make an adequate and complete explanation of the results obtained, adequately justifying the narrative they include. Table 2 is very relevant because it summarizes very well the practical implications of the study.

The study finds something very interesting about the stages that women go through, which are not detected in quantitative cross-sectional investigations, which is why the study is very relevant and necessary.

The discussion and conclusions of the study conform to the data found and do not make inappropriate generalizations.

For all the above, in my opinion the article is very relevant and necessary and deserves to be published.

Response 1: I am sincerely grateful for Your review, especially the observation that the phenomenon and qualitative approach of the study are relevant, and the study provides interesting and necessary results on the stages women go through after miscarriage. Thank you for delving into this article and providing valuable and encouraging insights from a qualitative research perspective.

Point 2: The method is clearly justified and is a suitable methodology for the purpose of the investigation. only, on Line 99 is missing the acronym for Thematic Analysis (TA). In the rest of the paragraph use the acronym.

Response 2: Thank you for this remark. This aspect has been corrected.
